# Toll-like Receptor Response to Human Immunodeficiency Virus Type 1 or Co-Infection with Hepatitis B or C Virus: An Overview

**DOI:** 10.3390/ijms24119624

**Published:** 2023-06-01

**Authors:** Mohammad Enamul Hoque Kayesh, Michinori Kohara, Kyoko Tsukiyama-Kohara

**Affiliations:** 1Department of Microbiology and Public Health, Faculty of Animal Science and Veterinary Medicine, Patuakhali Science and Technology University, Barishal 8210, Bangladesh; mehkayesh@pstu.ac.bd; 2Department of Microbiology and Cell Biology, Tokyo Metropolitan Institute of Medical Science, Tokyo 156-8506, Japan; kohara-mc@igakuken.or.jp; 3Transboundary Animal Diseases Centre, Joint Faculty of Veterinary Medicine, Kagoshima University, Kagoshima 890-0065, Japan

**Keywords:** toll-like receptor, TLR agonist, HIV-1, co-infection, HBV, HCV

## Abstract

Toll-like receptors (TLRs) are evolutionarily conserved pattern recognition receptors that play important roles in the early detection of pathogen-associated molecular patterns and shaping innate and adaptive immune responses, which may influence the consequences of infection. Similarly to other viral infections, human immunodeficiency virus type 1 (HIV-1) also modulates the host TLR response; therefore, a proper understanding of the response induced by human HIV-1 or co-infection with hepatitis B virus (HBV) or hepatitis C virus (HCV), due to the common mode of transmission of these viruses, is essential for understanding HIV-1 pathogenesis during mono- or co-infection with HBV or HCV, as well as for HIV-1 cure strategies. In this review, we discuss the host TLR response during HIV-1 infection and the innate immune evasion mechanisms adopted by HIV-1 for infection establishment. We also examine changes in the host TLR response during HIV-1 co-infection with HBV or HCV; however, this type of study is extremely scarce. Moreover, we discuss studies investigating TLR agonists as latency-reverting agents and immune stimulators towards new strategies for curing HIV. This understanding will help develop a new strategy for curing HIV-1 mono-infection or co-infection with HBV or HCV.

## 1. Introduction

Approximately 10% of human immunodeficiency virus (HIV)-infected individuals have chronic hepatitis B virus (HBV) co-infection in highly endemic areas of HBV infection [1,2,3]. In addition, approximately 2.3 million individuals are co-infected with HIV and the hepatitis C virus (HCV) worldwide [4]. However, specific molecular techniques are important for the accurate detection of particular viral agents with high sensitivity and high specificity [5,6]. During HIV co-infection, HBV infection shows an increased rate of persistence, higher HBV DNA levels, lower rates of hepatitis B e antigen loss, increased cirrhosis and liver-related mortality, and an increased risk of hepatocellular carcinoma (HCC) [2]. In addition, HIV/HBV co-infection accelerates the progression of liver cirrhosis and HCC compared to HIV or HBV mono-infected individuals [3,7]. Similarly, HIV co-infection also has a detrimental effect on the natural history of HCV, resulting in higher rates of HCV persistence following acute infection, higher viral loads, enhanced liver fibrosis, and development of HCC compared to HCV mono-infection [4,8,9]. Therefore, a proper understanding of the response of toll-like receptors (TLR) in HIV mono-infection or HIV/HBV and HIV/HCV co-infection is critical for developing HIV cure strategies in HIV mono-infection or HIV/HBV or HIV/HCV co-infection using TLR agonists as latency-reversing agents.

## 2. Toll-like Receptors

The innate immune system, a key component of host immunity, acts as the first line of defense against invading pathogens, including viruses [10]. The innate immune system detects pathogen-associated molecular patterns (PAMPs) or microbe-associated molecular patterns (MAMPs) and damage-associated molecular patterns (DAMPs) via germline-encoded pattern recognition receptors (PRRs) [11,12]. Toll-like receptors (TLRs) are key components of innate immunity and are evolutionarily conserved germline-encoded receptors [10]. TLRs can activate the transcription factor nuclear factor kappa B (NF-κB) and interferon regulatory factors (IRFs), which regulate the outcome of the innate immune response [13]. It is now becoming increasingly evident that TLRs are the most important family of PRRs and are among the key players in determining the outcome of microbial infections, including viruses [14,15,16,17,18]. Although TLRs play a vital role in early interactions with invading pathogens by detecting microbial PAMPs [19,20], a dysregulated TLR response may enhance immune-mediated pathology instead of providing protection [12,21,22]. Therefore, a proper understanding of TLR responses to pathogens, including viruses, is crucial.

TLRs are type I transmembrane proteins that contain a conserved N-terminal ectodomain of leucine-rich repeats, a single transmembrane domain, and a cytosolic Toll/interleukin (IL)-1 receptor (TIR) domain [10]. The TIR domain is common and identical among TLRs and the involvement of ILs binds them together as a unified structure [23]. TLRs are encoded by a large gene family containing 10 TLR members (TLR1–TLR10) in humans and 12 TLR members (TLR1–TLR9 and TLR11–TLR13) in mice [13]. They are localized on the cell surface or in intracellular compartments of the cell. TLRs localized on the cell surface include TLR1, TLR2, TLR4, TLR5, TLR6, and TLR10, and TLRs localized in intracellular compartments—including the endoplasmic reticulum, endosomes, lysosomes, or endolysosomes—comprise TLR3, TLR7, TLR8, and TLR9 [24,25]. Different TLRs are involved in recognizing various viral components. For example, TLR3, TLR7/8, and TLR9 are involved in the recognition of double-stranded viral RNA, single-stranded RNA, and DNA, respectively [26,27,28,29]. In contrast, TLR1, TLR2, TLR4, TLR5, and TLR6 are involved in the recognition of viral proteins [30].

## 3. Toll-like Receptors in Men and Women

Due to differences in sex, there may be a stronger or weaker immune response to viral infections [31,32]. TLRs exist in various immune cells and liver parenchymal cells, playing an important role in the host immune response [33]. The sex-related variability of the host immune response based on TLR response in viral infections has been observed [34]. It has been reported that the sensing of viral RNA by TLR7 is sex-biased, where greater expression of TLR7 is observed in female immune cells than in male immune cells [35,36]. TLR-mediated activation of pDCs in HIV-1 infection may induce higher immune activation in women than in men [37]. The TLR9 rs187084 C allele was found to be involved in spontaneous virus clearance in HCV infection in women, suggesting sex-specific effects of the TLR response in natural immunity against HCV infection [38].

## 4. Human Immunodeficiency Virus, Hepatitis B Virus, and Hepatitis C Virus

HIV, the etiologic agent of acquired immunodeficiency syndrome (AIDS) [39], remains one of the world’s most serious public health challenges. According to WHO estimates, 38.4 million [33.9–43.8 million] people live with HIV, and 650,000 people died of HIV-related illnesses in 2021 globally [40]. HIV infections can be categorized into two subgroups, type 1 and type 2. HIV type 1 (HIV-1) is the most prevalent cause of AIDS globally and has a higher virulence than HIV type 2 [41,42]. Therefore, most studies on pathogenesis and host immune responses have focused on HIV-1.

HIV-1 is an enveloped virus with a 9.6 kb positive-sense RNA genome that codes for three polyproteins (Gag, Pol, and Env) and six accessory proteins (Tat, Rev, Nef, Vpr, Vif, and Vpu) [43]. The HIV-1 ssRNA can be recognized by different PRRs, such as TLR7/TLR8 and retinoic acid-inducible gene I (RIG-I)/melanoma differentiation-associated protein 5 (MDA5), and may induce a profound antiviral inflammatory response [44,45,46] (Figure 1). HIV-1 replication intermediates, such as cDNA, ssDNA, DNA/RNA hybrids, and dsDNA, can be recognized by different host innate immune receptors, including DDX41, cGAS, and IFI16, thereby contributing to the antiviral response through the production of IFNs [47]. However, many reports have described the key role of innate immunity in the modulation of susceptibility to HIV infection, and TLR activation represents the first line of defense against invading pathogens [48,49,50,51]. However, TLR signaling may play a critical role in the outcome of HIV-1 infection (Figure 1). The rate of progression of HIV infection may vary among individuals [52]. Retained immune function and low levels of virological parameters indicate non-progressive HIV disease, whereas the loss of immune function and high levels of virus indicate rapid progression of the disease [53], suggesting a key role of the immune response in determining the rates of disease progression.

Due to the shared transmission routes of HBV and HCV, HIV/HBV or HIV/HCV co-infections are commonly observed [3,54]. HBV is an enveloped, circular, and partially double-stranded relaxed circular DNA (rcDNA) virus with a genome of approximately 3.2 kb that possesses four overlapping open reading frames that encode seven proteins, including polymerase, core, precore, three envelope/surface proteins (large, middle, and small), and X protein [55,56]. HCV is an enveloped, positive-sense, single-stranded RNA virus with a genome size of ~10 kb that encodes a large polyprotein that is cleaved into three structural (core, E1, and E2) and seven nonstructural proteins (p7, NS2, NS3, NS4A, NS4B, NS5A, and NS5B) by host and viral proteases [57]. Although HBV can behave as a stealth virus, several studies have reported the detection of HBV PAMPs using TLRs [56]. Notably, many reports have suggested a suppressed host TLR response in chronic HBV infection, and activation of the TLR response using TLR agonists as immunomodulators could improve HBV cure strategies [56]. HCV PAMPs are stronger inducers of the innate immune response compared to HBV, and differential expression of TLRs has been reported in HCV-infected patients [17,56,58,59]. Notably, a significant induction of intrahepatic TLR3, TLR7, and TLR8 mRNA was also observed in the livers of chronically infected tree shrews, squirrel-like non-primate small mammals of the Tupaiidae family that were found to be naturally susceptible to HCV infection [60,61,62]. Although further studies are warranted, it has been reported that TLR3 and TLR7 play a protective role in HCV infection, and TLR agonists show promise for their use in HCV infection as potential immunomodulators as well as vaccine adjuvants in HCV vaccine development [17]. Activation of TLR signaling by HBV and HCV viral PAMPs are depicted and reviewed in our recently published papers and can be read for further information [17,56].

## 5. TLR Response to Human Immunodeficiency Virus (HIV) Infection

Early innate viral recognition by host surface TLRs is important for a rapid antiviral response and for preventing infection. Although TLRs can be activated by viruses even before host cells become infected, the antiviral immune response is much more complicated where different cell types, including T cells, B cells, monocytes, NK cells, and neutrophils become involved, producing proinflammatory cytokines and chemokines for viral clearance or possibly promoting infection upon TLR engagement [63,64,65]. PAMP/MAMP-induced activation of TLR signaling suppresses the replication and spread of invading pathogens/microbes by the production of antimicrobial molecules such as type I interferon (IFN) and tumor necrosis factor (TNF)-α [66,67,68]. TLRs are important innate immune receptors that mediate immune activation during HIV-1 infection [69,70,71]. Through their specific ligands, TLRs initially activate the innate immune system, which helps shape the adaptive immune system. Differential regulation of TLR pathways has been observed in HIV-1 infection in monocytes, myeloid dendritic cells, and plasmacytoid dendritic cells, which are dependent on both individual receptors and cell types [69].

In a recent study of HIV infection, TLR2 and TLR4 were found to be differentially expressed [72]. In an earlier study, compared to healthy subjects, increased expression of TLR2 mRNA has been reported in monocytes isolated from HIV-1-infected patients [71]. TLR2 and TLR4 are involved in the regulation of the expression of proinflammatory cytokines in HIV-infected people. In ex vivo analysis, increased expression of TLR2 and TLR4 mRNA was observed in myeloid dendritic cells (mDCs) [71]. It has been shown that the triggering of TLR2 and TLR4 in response to productive HIV-1 infection of DCs and viral spreading to CD4+ T cells exert contrasting effects, where HIV-1 transmission into CD4+ T cells was increased by TLR2 stimulation but transmission to CD4+ T cells was reduced upon TLR4 triggering [73]. Another study reported that soluble TLR2 played a significant role in inhibiting cell-free HIV-1 infection, HIV-induced NF-κB activation, and inflammation in infants breastfeeding from their HIV-infected mother [74]. Increased expression of TLR6, TLR7, and TLR8 mRNA has been observed in chronic untreated HIV-1 infections [70]. TLR2, TLR3, and TLR4 mRNA expression was elevated in patients with advanced disease [70], suggesting sensitization of TLR signaling during HIV infection (Figure 2A). Prior addition of TLR3, TLR7, TLR8, and TLR9 agonists was shown to inhibit HIV-1 infection in peripheral blood mononuclear cells (PBMCs) (Figure 2B); TLR8 and TLR9 agonists were found to be more effective in blocking HIV-1 replication, along with the activation of other antiviral sensors [75].

Henrick et al. showed that the HIV-1 structural proteins p17, p24, and gp41 acted as viral PAMPs that initiated signaling through TLR2 and its heterodimers, resulting in increased immune activation via the NF-κB signaling pathway [76]. In an in vitro study with monocyte-derived macrophages, it has been shown that activation of TLR3 with poly (I:C), a synthetic ligand for TLR3, inhibits HIV replication by the induction of type I interferon-inducible anti-HIV cellular proteins, including APOBEC3G and tetherin [77]. Moreover, TLR3 activation induces CC chemokines and several cellular miRNAs, such as miRNA-28, -125b, -150, -223, and -382, which are known HIV restriction factors [77]. It has been reported that TLR3 agonists, poly (I:C), and bacterial ribosomal RNA could potently reactivate HIV in human microglial cells. Notably, Alvarez-Carbonell et al. reported a new mechanism of poly (I:C), where poly (I:C) did not activate NF-κB; instead virus induction was mediated by IRF3 [78].

It has been shown that TLR3 agonist poly (I:C) and CD40-targeting HIV-1 vaccine of a string of five highly conserved CD4+ and CD8+ T cell epitope-rich regions of HIV-1 Gag, Nef, and Pol fused to the C-terminus of a recombinant anti-human CD40 antibody (αCD40.HIV5pep) enhanced anti-HIV-1 immunity and reduced HIV-1 reservoirs in patients with suppressive combined antiretroviral therapy [79]. It has been reported that HIV-1 infection results in dysregulation of the TLR4 response to lipopolysaccharide (LPS) ex vivo [80]. Brenchley et al. demonstrated that microbial translocation was linked to increased plasma levels of LPS, the TLR4 ligand, which has been reported in chronic HIV-1 infection as well as in chronically SIV-infected rhesus macaques [81].

TLR7 recognizes RNA viruses, including HIV [82]. An early study reported the recognition of uridine-rich HIV-1 ssRNA by intracellular TLR7/8 [76]. It has been shown that the TLR7 agonist GS-9620 can activate HIV replication in PBMCs obtained from HIV-infected individuals on suppressive antiretroviral therapy [83]. Additionally, GS-9620 activated HIV-specific T cells and the antibody-mediated clearance of HIV-infected cells [83]. It was observed that imiquimod-based induction of TLR7 by macrophages could inhibit HIV infection by decreasing the expression of viral entry cellular factors, such as CD4 and CCR5 (Figure 2C) [84]. It has been reported that plasma CXCL13 levels increase with HIV disease progression [85]. HIV-1 induces CXCL13 production upon TLR7 activation (Figure 2D) [86]; however, the exact role of TLR7 in disease progression remains unknown. Although genetic polymorphisms provide a compelling link to the disease [87], the exact mechanism by which TLR polymorphisms affect HIV infection remains to be investigated. However, the association of different TLRs, including TLR1, TLR2, TLR3, TLR4, TLR6, TLR7, TLR8, and TLR9, in HIV infection has been observed [88,89]. TLR7 polymorphisms influence the susceptibility and progression of HIV-1 infection in Chinese men who have sex with men (MSM), suggesting an active role of TLR7 in HIV infection [82].

pDCs play an important role in host defense against viral infection, including HIV through TLR7 and TLR9 signaling [90,91,92]. HIV-1 encodes multiple uridine-rich TLR7/8 ligands, activates pDCs through TLR7/8, and produces IFN-I [29,44,93,94]. TLR agonists are reported to enhance HIV-specific T cell response, as well as inducing HIV-1 restriction factors and IFN-α production by pDCs [92]. During HIV chronic infection, there is reduced number of circulating pDCs, which is correlated with reduced CD4+ T cell counts in people living with HIV-1 (PLWH) [95,96]. TLR7-driven IFN-I production in pDCs is higher in women than in men, which could be due to the cell-intrinsic actions of estrogen and the X-chromosome complement [97]. Upon TLR7 activation, compared to uninfected controls, an increased expression of IFN-α and TNF-α by pDCs obtained from HIV-infected women under antiretroviral therapy has been reported [98]. In a previous study, the expression of TLR6 and TLR7 was found to be significantly correlated with plasma HIV-RNA load [70].

TLR8 recognizes RNA viruses including HIV [99,100]. The most frequent TLR8 polymorphism, TLR8 A1G (rs3764880), has been reported to confer protection against HIV progression, suggesting the role of TLR8 in infection [101]. Another study assessed TLR7 and eight gene polymorphisms associated with susceptibility to HIV-1 infection and found that TLR8 polymorphisms could affect HIV-1 infection [102]. TLR7 IVS2-151-A and TLR8 Met alleles affect plasma HIV viral loads [102].

Synthetic ligands of TLR8 also boost T cell receptor signaling, resulting in increased cytokine production and upregulation of surface activation markers, suggesting sensing of HIV-1 by TLR8 [100]. In addition, endosomal HIV-induced cytokine secretion from CD4+ T cells in a TLR8-specific manner has also been reported [100].

In a phase I study, MGN1703, a TLR9 agonist administered for 24 weeks, was found to be safe and to enhance innate as well as HIV-1-specific adaptive immunity in HIV-1+ individuals [103]. It has been reported that HIV-1 alone is a poor inducer of IFN-I response [104]; however, if supplemented with suboptimal levels of bacterial LPS, which may be due to microbial translocation, it may trigger increased production of IFN-I and upregulation of interferon-stimulated genes [105]. It has been reported that HIV infection can activate cyclic guanosine monophosphate-adenosine monophosphate (cGAMP) synthase (cGAS) to produce cGAMP, activating STING and induce type I IFNs and other cytokines [106]. LPS, in association with TLR2 or TLR4, may synergistically enhance IFN-I production by cGAMP, a secondary messenger of cGAS [105].

Until 2014, TLR10 remained without a defined ligand or function. Lee et al. first demonstrated a functional role of TLR10 in detecting influenza virus infection [107]. A role of TLR10 in HIV-1 infection has recently been described, whereby HIV-1 gp41 was recognized as a TLR10 ligand, leading to the induction of IL-8 and NF-κBα activation [108]. In vitro studies using TZMbl cells demonstrated that TLR10 overexpression could enhance HIV-1 infection and proviral DNA integration [108]. The interaction between TLRs and HIV-1 infection has been summarized in Table 1.

TLR response in HBV infection may modulate HBV-specific T and B cell responses, resulting in the termination of HBV infection [67]. Single nucleotide polymorphisms (SNPs) of TLRs may play an important role in HBV and HCV infection; however, the mechanisms remain to be fully understood [111]. The role of TLRs in HBV and HCV infection has been reviewed in detail in recently published papers [17,56]. TLR response-related data on HIV/HBV or HIV/HCV co-infection are scarce. Increased expression of TLRs in innate immune cells is observed in patients with a high HIV-1 load who are co-infected with opportunistic pathogens. Increased expression of TLR2 and TLR4 has been reported in the myeloid dendritic cells of HIV-1 patients co-infected with opportunistic infections (without HAART). Increased expression of TLR4 has been reported in plasmacytoid dendritic cells compared to both HIV-1 patients without opportunistic infections and healthy subjects [112].

## 6. Role of Interleukins and Other Cytokines in HIV-1 Infection

Virus infections can cause a proinflammatory response including expression of cytokines and chemokines [113]. Cytokines, signaling polypeptides, play an important role in immune activation, influencing viral pathogenesis [113,114]. HIV-1 infection may induce differential expression of cytokines and influence clinical outcomes [115]. A decreased secretion of Th1 cytokines such as IL-2 and antiviral IFN-γ, and an increased secretion of Th2 cytokines IL-4 and IL-10, and proinflammatory cytokines IL-1, IL-6, IL-8, and TNF-α have been reported in HIV-1 infection [115]. An increased levels of serum IL-6 has been reported in HIV chronic infection [116], and IL-6 was reported to enhance transcriptional levels of HIV-1 [117]. A previous study showed that HIV-1 Tat protein could activate TLR4 signaling and induce production of proinflammatory cytokines IL-6, IL-8, and TNF-α, and immunosuppressive cytokine IL-10 [118,119,120]. An increased circulating IL-7 level has been reported in HIV-1 infection, associating CD4+ T lymphopenia, and an increased IL-7 production was due to a homeostatic response to T cell depletion [121].

HIV-1 Nef protein can cause an induction of IL-15 induction and enhance viral replication [122]. A significant increase in IL-15 level has been reported in HIV-1 infected patients with viral loads >100,000 copies/ml compared to healthy subjects [123]. IL-15 has been shown to regulate the susceptibility of CD4+ T cells in HIV infection, and an inverse correlation between IL-15 levels and CD4+ T cell counts has been reported [123,124]. An increased serum level of IL-18 has been reported in the advanced and late stages of the disease, which significantly decreased after highly active antiretroviral therapy; however, in early stage there was no increase in IL-18 in HIV-1 infected patients [125]. Different cytokines such as IFN-alpha, IFN-beta, IL-10, IL-13, and IL-16 were reported to inhibit HIV-1 replication in T cells and/or monocyte-derived macrophages (MDM) [115]. Dysregulation of cytokines in HIV-1 infection is evident, and strategies targeting cytokines may open up new therapeutic intervention [126], requiring further investigation.

## 7. Inhibition of Innate Immune Response by HIV Infection

To infect susceptible hosts and ensure viral propagation towards establishing an infection, many viruses have developed strategies to evade or inhibit signaling through PRRs, including toll-like receptor pathways. HIV-1 can persistently infect humans and subvert their innate and adaptive immune systems, as HIV-1 primarily infects the CD4+ T lymphocytes and macrophages. The interaction between the trans-membrane domain (TMD) of the HIV-1 envelope protein and the TLR2 TMD may induce a reduced secretion of TNF-α, IL-6, and MCP-1 in macrophage [127]. It has been shown that HIV-1 gp120 can suppress the TLR9-mediated induction of proinflammatory cytokines and the expression of CD83, a marker of DC activation, resulting in a decreased ability of pDCs to secrete antiviral and inflammatory factors [128].

Sterile-motif/histidine-aspartate domain-containing protein 1 (SAMHD1), a host restriction factor possessing dNTP triphosphohydrolase activity, has been reported to restrict HIV-1 infection in monocyte-derived macrophages (MDMs) by reducing the intracellular dNTP pool [129,130]. SAMHD1 was also reported to inhibit an efficient viral DNA synthesis in non-cycling resting CD4+ T cells [131]. However, the HIV-1 accessory protein Vpx counteracts this restriction by directing SAMHD1 toward proteasomal degradation [132,133]. In addition, SAMHD1 suppresses the innate immune response to viral infections, including HIV-1 infection, and inflammatory stimuli by inhibiting NF-κB activation and type I IFN induction [134]. Moreover, HIV-1 capsids may inhibit viral DNA sensing via cGAS [135].

APOBEC3G, a cellular restriction factor, inhibits HIV-1 replication by inhibiting the elongation of HIV-1 reverse transcripts [136]. However, the viral protein Vif helps evade the host antiviral factor APOBEC3G by binding and inducing its degradation in virus-producing cells [137,138,139,140]. The inhibition of HIV-1 release mediated by tetherin, a cellular factor, can be antagonized by HIV-1 Vpu [141,142,143,144]. It has been shown that HIV-1 Vpr can inhibit IRF3 and NF-κB nuclear transport by interacting with karyopherins [145].

However, different accessory proteins may induce differential IFN response, including a potentiated type 1 IFN response by Vpr, and a suppressed IFN response by Vpu was observed in HIV-1 infected CD4^+^ T cells [47]. The antiviral mechanisms adopted by HIV-1 largely depend on HIV-1 accessory proteins and their ability to degrade/inhibit/downregulate host antiviral targets [47]. The inhibition of host response by HIV-1 proteins is summarized in Table 2. For information on the inhibition/avoidance of innate immune response by HBV or HCV infection, other recently published papers can be studied [17,56].

## 8. Enhanced Pathogenesis of HIV/HBV or HIV/HCV Co-Infection

During HIV-1 infection, the immune system fails to completely eliminate the infecting virus, and complete inhibition of viral replication by the immune system is achieved in only a very small fraction of infected individuals [157]. Moreover, the elimination of viral infection or the complete prevention of viral replication cannot be achieved with current antiretroviral (ART) therapies [158]. Both reduction in the amount of virus persisting on antiretroviral therapy and enhanced anti-HIV immune surveillance are critical for curing HIV infection [159]. Different interventions, including the reversion of HIV latency by apoptosis-promoting agents, non-histone deacetylase inhibitor compounds, and immunotherapies, such as immune checkpoint inhibitors and TLR agonists, have been investigated to enhance antiviral immunity in curing HIV infection [159].

HIV infection can disrupt the gut mucosal barrier, resulting in microbial translocation and increased exposure to microbial products that act as TLR agonists [160]. Recent studies suggest that TLR signaling is an underlying mechanism for persistent immune activation, which is linked to progressive HIV-1 disease [161,162]. Further details on the immunopathogenesis of HIV-1 infection can be found in previously published reviews [163,164].

Continuous surveillance is critical for tracking viral diversity, as there is an increase in recombinant viruses, resulting in co-infection and superinfection by divergent HIV strains [41]. Individuals infected with HIV may show a higher tendency to develop chronic HBV infections [165]. Hepatocytes express CCR5 and CXCR4—two co-receptors for HIV entry—and multiple cells in the liver can be infected with HIV, which is associated with liver dysfunction [99,166,167]. HIV infection may induce increased levels of HBV DNA in chronic HBV infection compared with HBV mono-infection [7]. HIV/HBV co-infection enhances HBV-related hepatotoxicity mediated by an immune response [2]. Higher liver-related mortality was observed in HIV/HBV co-infection with lower CD4+ T cell counts than in HIV or HBV mono-infection [168]. An approximately five- to six-fold higher risk of HCC has been reported in HBV/HIV co-infected individuals [169,170].

Increased liver fibrosis is observed in HIV-1/HCV co-infection; however, the underlying mechanism is complex. HIV-1 enteropathy may cause increased microbial translocation and systemic immune activation, which may promote liver fibrosis either by direct interaction with Kupffer cells and hepatic stellate cells or indirectly via the induction of systemic immune activation and activation-induced apoptotic cell death [171]. Moreover, an increased risk of liver fibrosis in HIV-1/HCV co-infection may be associated with reduced CD4 + T cell counts, elevated lipopolysaccharide levels, and/or the depletion of hepatic Kupffer cells [171]. Although oral direct-acting antivirals have become very effective and can cure almost all treated patients infected with HCV, as there is no protective HCV immunity, PLWH with high-risk behaviors may experience HCV reinfections [172].

It has been reported that HIV co-infection may increase profibrogenic cytokine production with enhanced oxidative stress and hepatocyte apoptosis, which may be further augmented by increased microbial translocation of HIV [173]. Another study reported that HIV/HCV co-infection significantly increased epimorphin expression, which may result in increased proliferation and invasion capabilities of hepatic stellate cells [174]. Epimorphin induces the expression of profibrogenic tissue inhibitor of metalloproteinase 1 (TIMP1) in an extracellular signal-regulated kinase (ERK)-dependent manner [174].

HCV infection can induce transforming growth factor-beta1 (TGF-β1) expression in hepatocytes, which has been reported to be further enhanced by HIV co-infection [175]. HIV-1 Tat could activate HCV replication by upregulating interferon gamma-induced protein-10 (IP-10) production [176]. In addition, increased expression of IP-10 mRNA in peripheral blood mononuclear cells has been reported in HIV-1/HCV co-infection. Higher HCV RNA levels were found in the PBMCs of patients with HIV-1/HCV co-infection than in those with HCV mono-infection [176]. Viral infections, including HIV, HBV, and HCV, have been reported to cause the upregulation of programmed cell death protein (PD-1) and its ligands PD-L1 and PD-L2, indicating that PD-1/PD-L1 are key players in the pathogenic process of viruses [177]. Notably, the induction of anti-HBs titers in HBV vaccines is influenced by HIV infection, with a high percentage (90% to 95%) of healthy adults inducing protective anti-HBs titers after standard doses of HBV vaccines; however, during HIV co-infection, only 17.5% to 71% could retain protective anti-HBs [178].

## 9. TLR Agonists and Antagonists in the Cure of HIV Infection

The presence of latent and persistent viral reservoirs in transcriptionally silent immune cells remains a challenge to a successful HIV cure [179]. The persistent HIV-1 reservoir is predominantly found in long-lived memory CD4+ T cells, where the provirus remains transcriptionally silent and undetected by the host immune system [180,181]. Although ART can suppress HIV-1 replication to undetectable levels, it cannot eliminate latent and persistent viral reservoirs and the use of ART may result in multiple side effects including the development of multidrug-resistant escape mutants [182]. Therefore, it is critical to develop a strategy towards eliminating latently infected cells persisting in people with HIV on ART and identifying an ideal latency-reversing agent (LRA) that can induce viral reactivation, leading to immune cell recognition and the elimination of latently infected cells, which may improve HIV cure strategies [183]. Notably, many studies have indicated that TLR agonists are potential immunomodulators of the host immune response and influence the outcome of chronic HBV and HCV infections, suggesting the merit of using TLR agonists in curing chronic HIV-1 infection [17,56,183].

TLR agonists have the potential to reactivate latent HIV, induce immune activation, and promote an antiviral response by improving HIV-specific cytotoxic CD8+ T cell immunity, making TLR agonists unique LRA [179,183,184,185,186]. Pam3CSK4, a TLR1/2 agonist, has been shown to induce viral reactivation in a Tat-dependent manner and in the absence of T cell activation and/or proliferation [185]. TLR7 agonists have been reported to facilitate the reduction of viral reservoirs in a subset of SIV-infected rhesus macaques [187]. In a phase 1b study, vesatolimod, an oral TLR7 agonist, was found to be well tolerated, and immune stimulation was observed, providing a rationale for future combination trials in people living with HIV [188]. Dual TLR2/7 agonists have been shown to have a greater ability to induce the production of soluble factors for the reactivation of latent HIV than the combination of a single TLR2 and TLR7 agonist [189]. Another recent study of the synthesized TLR7/TLR8 agonist 159 pyrido [3,2-d] pyrimidine and pyridine-2-amine-based derivatives reported a high potential for activating HIV-1 latent reservoirs in cell lines and PBMCs of patients with persistent and durable virologic controls [182]. A previous study showed that the TLR9 agonist MGN1703 enhanced HIV-1 transcription and cytotoxic NK cell activation, which could contribute to HIV-1 eradication therapy [190].

The stimulation of TLR3, TLR4, TLR7, and TLR7/8 by the agonists poly (I:C), LPS, imiquimod, and ssRNA40, respectively, in PBMCs isolated from HIV-1-exposed seronegative (ESN) individuals induced a robust release of immunological factors in ESN individuals and showed a protective phenotype against HIV-1 [191]. The combination strategy using TLR agonist-directed innate effector functions and broadly neutralizing antibodies (bNAbs) may help enhance the elimination of HIV-1 infected cells [192,193]. Although there is hope for the success of curing HIV-1 infection using TLR agonists, further research is required to obtain clinical data and select the most suitable candidates for future use. Moreover, studies on the use of TLR agonists in HBV/HIV or HCV/HIV co-infection are scarce. However, a study was initiated to investigate the effect of the oral TLR8 agonist Selgantolimod on HBsAg in participants with both chronic HBV and HIV infections (ClinicalTrials.gov Identifier: NCT05551273), which may provide new data.

TLRs may act as a double-edged sword, and its double-sided effects have been reported in viral infections [21,194,195]. TLR ligand-induced chronic immune activation due to HIV-1 infection or microbial translocation from the gut, resulting in the chronic production of proinflammatory cytokines in HIV-1–infected individuals, may progress the disease. An abnormal activation of TLR7 may stimulate HIV infection [196]; therefore, the reduction or inhibition of chronic immune activation via TLR stimulation could be investigated for a potential target in reducing HIV-related immunopathology [197]. CXCR4 acts as a co-receptor for CXCR4-tropic HIV entry to CD4+ T cell [198]. CXCR4 has recently been shown to be involved in an additional regulatory function by directly inhibiting the induction of TLR signaling and resulting in a direct influence on IFN-I levels [199,200]. An engagement of CXCR4 can be used for inhibiting pDC activation and the control of TLR7-mediated IFN-I signaling [201,202]. Antibody-based treatment strategies, such as the use of anti-CCR5 antibodies targeting coreceptor CCR5, could be considered as alternative approaches for curing HIV-1 infection [203]. Although much knowledge has been gathered, further investigations are warranted for the clinical use of TLR agonists and antagonists in the cure of HIV-1 infection. The role of TLR agonists and antagonists in the cure of HIV-1 infection has been summarized in Table 3.

## 10. Conclusions

TLRs can produce redundant responses such as inflammatory or antiviral responses; however, there are differences in outcomes, largely influenced by the roles of ligands and tissue-dependent TLR expression [205,206,207]. The findings of different studies indicate that TLRs are active participants and key regulators of the antiretroviral immune regulation. However, the interaction between HIV-1 and the TLR system is full of complexity, and the dual role of the TLR system affecting viral replication and immune cell functioning requires detailed investigation to understand the specific roles of individual TLRs and their agonists and antagonists in both HIV-1 pathogenesis and protection. Robust immune surveillance is required for eradicating persistent HIV [208]. TLR agonists can be used in reactivating latent HIV-1 reservoirs to make them visible to the immune system. Although further studies are warranted, the use of TLR agonists and antagonists in treating HIV infection may become promising with future research works; however, there are very limited data for HIV/HBV or HIV/HCV co-infection, requiring further studies for their clinical use.

## Figures and Tables

**Figure 1 ijms-24-09624-f001:**
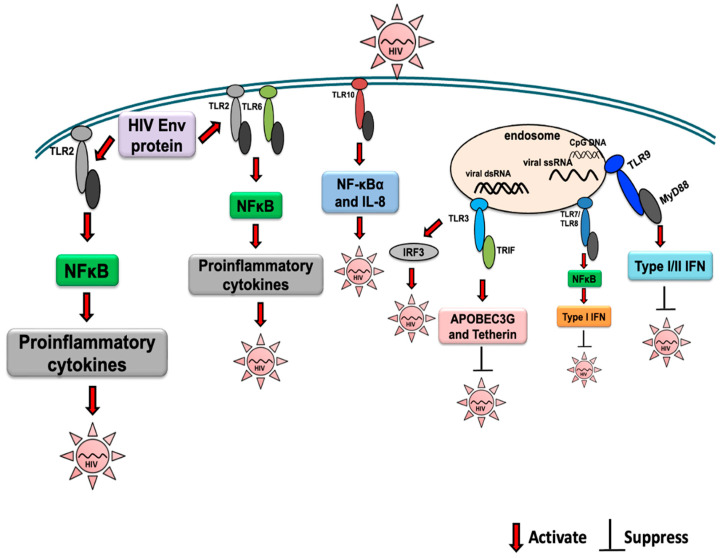
Activation of TLR signaling during HIV infection and the effect on viral replication are indicated. The activation and suppression of signaling is indicated by arrows.

**Figure 2 ijms-24-09624-f002:**
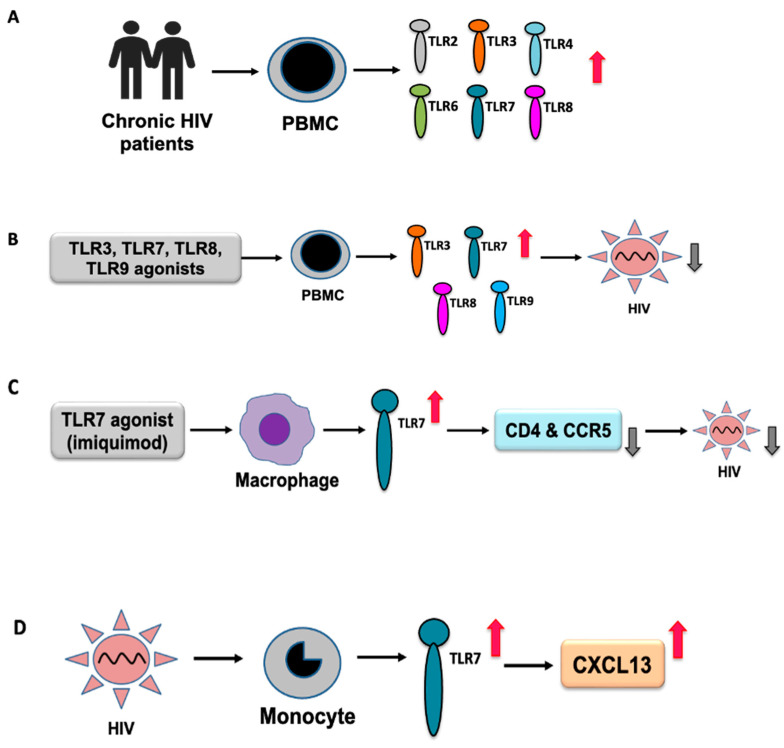
An overview of differential TLR expression in HIV infection. (**A**) Increased expression of TLRs in PBMCs obtained from patients with chronic HIV infection. (**B**) Induction of TLR responses using TLR agonists for TLR3, 7, 8, and 9 inhibited HIV replication in PBMCs. (**C**) Induction of the TLR7 response in macrophages by imiquimod, a TLR7 agonist, inhibited HIV replication by inhibiting CD4 and CCR5. (**D**) HIV-induced production of CXCL13 through TLR7 activation in monocytes.

**Table 1 ijms-24-09624-t001:** Interaction between TLRs and HIV-1 infection.

HIV-1 Infection System	TLR Ligands	Activation of TLR	Effects on Host Response	Outcome	Reference
Monocyte-derived macrophage (MDM) and PBMCs	HIV proteins	TLR2, TLR4	Increases production of proinflammatory cytokines	Promotes HIV-1 replication	[71]
MDM	Poly (I:C), a TLR3 ligand	TLR3	Induces anti-HIV cellular proteins APOBEC3G and tetherin	Inhibits HIV replication	[77]
PBMCs	TLR3, TLR7, TLR8 and TLR9 agonists	TLR3, TLR7, TLR8, TLR9	Induces type I and type II IFNs and ISGs	Reduces viral replication	[75]
T cells and TZMbl-2 cells (stably expressing TLR2)	HIV-1 structural proteins p17, p24, and gp41	TLR2 or TLR2/1 or TLR2/6 heterodimers	Activates NF-κB signaling pathway	Enhances proviral DNA	[76]
Microglial cells	Poly (I:C)	TLR3	Selective induction of IRF3	Reactivates HIV transcription	[78]
PBMCs	GS-9620, a TLR7 agonist	TLR7	Enhances HIV-specific cellular immunity	Enhances cytokines-mediated HIV replication	[83]
Macrophage	Imiquimod, a TLR7 agonist	TLR7	Reduces viral entry factors such as CD4 and CCR5	Inhibits HIV replication	[84]
DCs	Free HIV-1	TLR8	Activation of IRF1, p38, ERK, PI3K, and NF-κB pathways	Enhances infection	[109]
DCs and macrophages	HIV ssRNA	TLR7, TLR8	Enhances immune activation and secretion of cytokines	May inhibit viral replication	[29,44]
Macrophage	CpG ODN 2216	TLR9	Increases expression of IFN-α, IRF-7, MyD88, and myxovirus resistance gene A	Inhibits HIV replication	[110]
TZMbl cells	HIV-1 gp41	TLR10	Induces IL-8 and NF-κBα activation	Enhances HIV-1 infection	[108]

**Table 2 ijms-24-09624-t002:** Inhibition of host innate immune response by HIV-1.

HIV Protein	Targeting Protein/Signaling Pathway	Effects on Host Immunity	Effects on Virus	Reference
HIV-1 Env	Binds TLR2	Inhibits secretion of pro-inflammatory cytokines	-	[127]
HIV-1 Vpr	Interacts with karyopherins	Inhibits IRF3 and NF-κB nuclear transport	Enhances virus replication	[145]
HIV-1 Vif	Degrades APOBEC3G	Inhibits host antiviral activity of APOBEC3G	Overcomes inhibition of virus replication	[138,140,146,147]
HIV-1 Vpu	Degrades/downregulates tetherin/BST-2	Inhibits host antiviral activity of tetherin	Facilitates virus release	[141,142,148]
HIV-1 Vpu	Host restriction factors	Inhibits ISGylation of cellular machinery	Enhances late-stage virus replication	[149]
HIV-1 Vpu	Inhibits NF-κB activation	Reduces expression of restriction factors and IFNs release	May promote viral replication	[150,151,152]
HIV-1 Vpx	SAMHD1	Degrades SAMHD1	Facilitates reverse transcription	[132,133]
HIV-1 Nef	Activates IL-15 synthesis	Modulates cytokine response	Enhances HIV-1 replication	[122]
HIV-1 Nef	SERINC5 and SERINC3 proteins	Inhibits host’s retroviral restriction of SERINC5 and SERINC3	Promotes HIV-1 infectivity	[153,154]
HIV-1 Nef	CD36, a scavenger receptor	Downregulates CD36 expression	Enhances opportunistic infection	[155]
HIV-1	Cellular 2’-O-methyltransferase	Avoids innate immune recognition	Acquires 2′-O-methylation at multiple distinct sites of HIV-1 RNA	[156]

**Table 3 ijms-24-09624-t003:** Immunomodulatory effects of TLR agonists and antagonists in the cure of chronic HIV infection.

Compound (TLR Agonist or Antagonist)	Sponsor	Target TLR	Clinical Phase	Effects on Host Immunity	Effects on Virus	Reference/ClinicalTrials.gov ID
Pam3CSK4	-	TLR1/2	-	Transcription factors NFκB, NFAT, and AP-1 cooperate to induce viral reactivation	Enhances virus replication	[185]
Vesatolimod	-	TLR7	Phase 1b	Safe and well tolerated; enhances immune stimulation	No significant effect on plasma viral load	[188]
GS-9620 (Vesatolimod)	Gilead Sciences	TLR7	Phase 2	No clinical data available	No clinical data available	NCT05281510
MGN1703 (Lefitolimod)	University of Aarhus	TLR9 agonists	Phase 2	No clinical data available	No clinical data available	NCT03837756
MGN1703 (Lefitolimod)		TLR9	Phase 1b/2a	Enhances innate immunity; increases transcription of IFNAR1; no concomitant general inflammatory response in the intestines; reduction of HIV-1 viral reservoir	Decreases integrated HIV-1 DNA	[204]
MGN1703	University of Aarhus	TLR9	Phase 1 Phase 2	Safe and well-tolerated; enhances both innate and adaptive immunity; increases HIV-1-specific T cell responses; enhances cytotoxic NK cell activation	Enhances HIV-1 replication	[103,190]

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
