# Peer review of "Toll-like Receptor Response to Human Immunodeficiency Virus Type 1 or Co-Infection with Hepatitis B or C Virus: An Overview"

_ijms, 2023, doi:10.3390/ijms24119624_

Round 1

Reviewer 1 Report

Dear Authors

Thank you for your manuscript submission. The topic of your manuscript is interesting; however, a Major Revision is needed as below:

1. Please do read and add the following papers to References section of the manuscript to make Introduction section of the manuscript fruitful:

Nucleic Acid-based approaches for detection of viral hepatitis. Jundishapur J Microbiol. 2014 Dec 10;8(1):e17449. doi: 10.5812/jjm.17449. PMID: 25789132; PMCID: PMC4350052.

DNA microarray technology in HBV genotyping. Minerva Med. 2017 Oct;108(5):473-476. doi: 10.23736/S0026-4806.17.05059-5. PMID: 28728341.

2. Please do read and add the following papers to References section of the manuscript to make the subtitle of "2. Toll-like receptors" fruitful.

Toll-Like Receptors (TLRs) in Health and Disease: An Overview. Handb Exp Pharmacol. 2022;276:1-21. doi: 10.1007/164_2021_568. PMID: 35091824.

The Interleukin-1 (IL-1) Superfamily Cytokines and Their Single Nucleotide Polymorphisms (SNPs). J Immunol Res. 2022 Mar 26;2022:2054431. doi: 10.1155/2022/2054431. PMID: 35378905; PMCID: PMC8976653.

3. As this manuscript is a Review, it is recommended to add a subtitle as "Toll-like receptors in men and women". As you know some TLRs functions and genetics may be stronger or weaker in males and females.

4. It is recommended to add a table to show the level of different TLRs and their association in patients with HIV, HBV, HCV in separate and patients with HIV or HIV together with HBV or HCV or HIV and HBV and HCV.

5. It is recommended to add the association with TLRs with different structural and non-structural proteins in viruses of HIV, HBV and HCV. A figure should be prepared in this regard.

6. It is recommended to add a figure to show the processes mentioned in subtitle "Human immunodeficiency virus, hepatitis B virus, and hepatitis C virus"

7. It is recommended to read and add the following papers to References section to make subtitle of "TLR response to human immunodeficiency virus (HIV) infection" fruitful:

Contribution of T- and B-cell intrinsic toll-like receptors to the adaptive immune response in viral infectious diseases. Cell Mol Life Sci. 2022 Oct 12;79(11):547. doi: 10.1007/s00018-022-04582-x. PMID: 36224474; PMCID: PMC9555683.

Scratching the Surface Takes a Toll: Immune Recognition of Viral Proteins by Surface Toll-like Receptors. Viruses. 2022 Dec 24;15(1):52. doi: 10.3390/v15010052. PMID: 36680092; PMCID: PMC9863796.

Role of TLRs in HIV-1 Infection and Potential of TLR Agonists in HIV-1 Vaccine Development and Treatment Strategies. Pathogens. 2023 Jan 5;12(1):92. doi: 10.3390/pathogens12010092. PMID: 36678440; PMCID: PMC9866513.

Identification of Molecular Mechanisms Involved in Viral Infection Progression Based on Text Mining: Case Study for HIV Infection. Int J Mol Sci. 2023 Jan 11;24(2):1465. doi: 10.3390/ijms24021465. PMID: 36674980; PMCID: PMC9862153.

Hepatitis C Virus Infection and Intrinsic Disorder in the Signaling Pathways Induced by Toll-Like Receptors. Biology (Basel). 2022 Jul 21;11(7):1091. doi: 10.3390/biology11071091. PMID: 36101469; PMCID: PMC9312352.

8. What is the role of interleukins and other cytokines in this regard? Please do add a subtitle in this regard.

9. Please do revise the subtitle of "TLR agonists in the cure of HIV infection

as below:

TLR agonists and antagonists in the cure of HIV infection

In this regard please do read and add the following papers to References section of the manuscript to have effective section:

The role of latency reversal in HIV cure strategies. J Med Primatol. 2022;51:278-283. doi: 10.1111/jmp.12613

Recent Advances on Small-Molecule Antagonists Targeting TLR7. Molecules. 2023 Jan 7;28(2):634. doi: 10.3390/molecules28020634. PMID: 36677692; PMCID: PMC9865772.

Immunotherapy with Cell-Based Biological Drugs to Cure HIV-1 Infection. Cells. 2021 Dec 28;11(1):77. doi: 10.3390/cells11010077. PMID: 35011639; PMCID: PMC8750418.

Targeting Cellular and Tissue HIV Reservoirs With Toll-Like Receptor Agonists. Front Immunol. 2019 Oct 15;10:2450. doi: 10.3389/fimmu.2019.02450. PMID: 31681325; PMCID: PMC6804373.

CXCR4 as a novel target in immunology: moving away from typical antagonists. Future Drug Discov. 2022 Jun;4(2):FDD77. doi: 10.4155/fdd-2022-0007. Epub 2022 Jul 19. PMID: 35875591; PMCID: PMC9298491.

10. It is recommended to add a table containing all the related agonists and antagonist together with associated effects.

acceptable

Author Response

Thank you for your manuscript submission. The topic of your manuscript is interesting; however, a Major Revision is needed as below:

  1. Please do read and add the following papers to References section of the manuscript to make Introduction section of the manuscript fruitful:

Nucleic Acid-based approaches for detection of viral hepatitis. Jundishapur J Microbiol. 2014 Dec 10;8(1):e17449. doi: 10.5812/jjm.17449. PMID: 25789132; PMCID: PMC4350052.

DNA microarray technology in HBV genotyping. Minerva Med. 2017 Oct;108(5):473-476. doi: 10.23736/S0026-4806.17.05059-5. PMID: 28728341.

Response: The suggested papers have been added (ref. 5,6)

  1. Please do read and add the following papers to References section of the manuscript to make the subtitle of "2. Toll-like receptors" fruitful.

Toll-Like Receptors (TLRs) in Health and Disease: An Overview. Handb Exp Pharmacol. 2022;276:1-21. doi: 10.1007/164_2021_568. PMID: 35091824.

The Interleukin-1 (IL-1) Superfamily Cytokines and Their Single Nucleotide Polymorphisms (SNPs). J Immunol Res. 2022 Mar 26;2022:2054431. doi: 10.1155/2022/2054431. PMID: 35378905; PMCID: PMC8976653.

Response: The suggested papers have been added (ref. 12, 23)

  1. As this manuscript is a Review, it is recommended to add a subtitle as "Toll-like receptors in men and women". As you know some TLRs functions and genetics may be stronger or weaker in males and females.

Response: In line with reviewer suggestion, we have added a subsection titled “Toll-like receptors in men and women” (page 2, lines 77-87)

  1. It is recommended to add a table to show the level of different TLRs and their association in patients with HIV, HBV, HCV in separate and patients with HIV or HIV together with HBV or HCV or HIV and HBV and HCV.

Response: In response to reviewer comments, we have added the association of TLR and HIV-1 in Table 1, and for the association of TLRs in HBV and HCV infection we suggest our recently published papers (Kayesh et al., 2021, PMID: 34638802; Kayesh et al., 2022, PMID: 35628287).

  1. It is recommended to add the association with TLRs with different structural and non-structural proteins in viruses of HIV, HBV and HCV. A figure should be prepared in this regard.

Response: In response to reviewer comments, we have prepared a figure indicating viral PAMPs-mediated activation of TLR signaling during HIV infection and their effects on host-virus interaction (newly added Figure 1), and for HBV and HCV infection and for their effects on host-virus interaction recently published papers are suggested (Kayesh et al., 2021, PMID: 34638802; Kayesh et al., 2022, PMID: 35628287).

  1. It is recommended to add a figure to show the processes mentioned in subtitle "Human immunodeficiency virus, hepatitis B virus, and hepatitis C virus"

Response: In response to reviewer comments, we have included the processes involved in HIV infection in newly added Figure 1, and for the processes involved in HBV, and HCV sensing by TLRs and downstream signaling are depicted in other papers and referred (Kayesh et al., 2021, PMID: 34638802; Kayesh et al., 2022, PMID: 35628287).

  1. It is recommended to read and add the following papers to References section to make subtitle of "TLR response to human immunodeficiency virus (HIV) infection" fruitful:

Contribution of T- and B-cell intrinsic toll-like receptors to the adaptive immune response in viral infectious diseases. Cell Mol Life Sci. 2022 Oct 12;79(11):547. doi: 10.1007/s00018-022-04582-x. PMID: 36224474; PMCID: PMC9555683.

Scratching the Surface Takes a Toll: Immune Recognition of Viral Proteins by Surface Toll-like Receptors. Viruses. 2022 Dec 24;15(1):52. doi: 10.3390/v15010052. PMID: 36680092; PMCID: PMC9863796.

Role of TLRs in HIV-1 Infection and Potential of TLR Agonists in HIV-1 Vaccine Development and Treatment Strategies. Pathogens. 2023 Jan 5;12(1):92. doi: 10.3390/pathogens12010092. PMID: 36678440; PMCID: PMC9866513.

Identification of Molecular Mechanisms Involved in Viral Infection Progression Based on Text Mining: Case Study for HIV Infection. Int J Mol Sci. 2023 Jan 11;24(2):1465. doi: 10.3390/ijms24021465. PMID: 36674980; PMCID: PMC9862153.

Hepatitis C Virus Infection and Intrinsic Disorder in the Signaling Pathways Induced by Toll-Like Receptors. Biology (Basel). 2022 Jul 21;11(7):1091. doi: 10.3390/biology11071091. PMID: 36101469; PMCID: PMC9312352.

Response: The suggested and other relevant papers have been cited (ref. 63-68, 87-96, 107-108, 111)

  1. What is the role of interleukins and other cytokines in this regard? Please do add a subtitle in this regard.

Response: In response to reviewer comments, we have added a subtitle and described the role of interleukins and other cytokines in HIV infection (page 8, line 280-306).

  1. Please do revise the subtitle of "TLR agonists in the cure of HIV infection" 

as below:

TLR agonists and antagonists in the cure of HIV infection

In this regard please do read and add the following papers to References section of the manuscript to have effective section:

The role of latency reversal in HIV cure strategies. J Med Primatol. 2022;51:278-283. doi: 10.1111/jmp.12613

Recent Advances on Small-Molecule Antagonists Targeting TLR7. Molecules. 2023 Jan 7;28(2):634. doi: 10.3390/molecules28020634. PMID: 36677692; PMCID: PMC9865772.

Immunotherapy with Cell-Based Biological Drugs to Cure HIV-1 Infection. Cells. 2021 Dec 28;11(1):77. doi: 10.3390/cells11010077. PMID: 35011639; PMCID: PMC8750418.

Targeting Cellular and Tissue HIV Reservoirs With Toll-Like Receptor Agonists. Front Immunol. 2019 Oct 15;10:2450. doi: 10.3389/fimmu.2019.02450. PMID: 31681325; PMCID: PMC6804373.

CXCR4 as a novel target in immunology: moving away from typical antagonists. Future Drug Discov. 2022 Jun;4(2):FDD77. doi: 10.4155/fdd-2022-0007. Epub 2022 Jul 19. PMID: 35875591; PMCID: PMC9298491.

Response: As per reviewer comments we have revised the subtitle as “TLR agonists and antagonists in the cure of HIV infection” and also added the suggested and relevant papers (ref. 21, 182,185,194-203).

  1. It is recommended to add a table containing all the related agonists and antagonist together with associated effects.

Response: As per reviewer comments, the effects of using TLR agonists and antagonists in the cure of HIV-1 infection have been included in newly added Table 3.

Reviewer 2 Report

In this review, authors discuss the host TLR response during HIV-1 infection and the innate immune evasion mechanisms adopted by HIV-2 for infection establishment. They also examined changes in the host TLR response during HIV-1 co-infection with HBV or HCV. This review is very interesting, but TLRs in HIV infection have been extensively described, while are very brief to HBV/HCV infection.

I suggest reviewing the manuscript and including the following information:

Consider the TLR agonists for targeting HIV-1, o coinfection HIV-1/HBV or HIV-1/HCV.

I suggest to describe using agonists for inmunotherapy in chronic HIV-1 infection or co-infection HIV-1/HBV or HIV-1/HCV. 

I suggest including a table with the mechanisms employed by HIV-1, o co-infection HIV-1/HBV or HIV-1/HCV to counteract the TLR system. Here you can describe the experimental model, mechanism HIV-1/HBV or HIV-1/HCV protein involved, and functional outcome.

The mechanisms of TLR-mediated inhibition of HBV through activation of innate immunity and modulation of adaptive immunity. Consider a Table with the experimental model, mechanism TLRs involved, and antiviral effects

In addition, the authors should describe (in a Table) the using TLR agonists for immunotherapies of chronic HIV-1 infection or co-infection HIV-1/HBV or HIV-1/HCV. 

Author Response

In this review, authors discuss the host TLR response during HIV-1 infection and the innate immune evasion mechanisms adopted by HIV-2 for infection establishment. They also examined changes in the host TLR response during HIV-1 co-infection with HBV or HCV. This review is very interesting, but TLRs in HIV infection have been extensively described, while are very brief to HBV/HCV infection.

Response: We thank reviewer for his sincere comments. In response to reviewer comment, we have referred papers for detailed TLR response in HBV and HCV infection (page 5, line 265-266).

I suggest reviewing the manuscript and including the following information:

Consider the TLR agonists for targeting HIV-1, o coinfection HIV-1/HBV or HIV-1/HCV.

Response: In line with reviewer comments, we have added the information for TLR agonists targeting HIV-1 infection. However, there is scarcity of study of HIV/HBV or HIV/HCV coinfection and already we mentioned in the text (lines 266-267).

I suggest to describe using agonists for immunotherapy in chronic HIV-1 infection or co-infection HIV-1/HBV or HIV-1/HCV. 

I suggest including a table with the mechanisms employed by HIV-1, o co-infection HIV-1/HBV or HIV-1/HCV to counteract the TLR system. Here you can describe the experimental model, mechanism HIV-1/HBV or HIV-1/HCV protein involved, and functional outcome.

Response: As per reviewer comments, we have included the inhibition or avoidance of host response by HIV-1 in a newly added Table 2. As mentioned above there is scarcity of study for information on HIV co-infection with HBV or HCV.

The mechanisms of TLR-mediated inhibition of HBV through activation of innate immunity and modulation of adaptive immunity. Consider a Table with the experimental model, mechanism TLRs involved, and antiviral effects

Response: We thank the reviewer for the comments. In response to reviewer comments, for related information we have referred other relevant review paper (Kayesh et al., 2021, PMID: 34638802).

In addition, the authors should describe (in a Table) the using TLR agonists for immunotherapies of chronic HIV-1 infection or co-infection HIV-1/HBV or HIV-1/HCV. 

Response: As per reviewer comments, the effects of using TLR agonists and antagonists in the cure of HIV-1 infection have been included in Table 3. And as already mentioned, there is scarcity of HIV coinfection with HBV/HCV, and the available information has been included. 

Reviewer 3 Report

Article titled: ''Toll-like Receptor Response To Human Immunodeficiency Virus Type 1 or Co-infection With Hepatitis B or C virus Infection: An Overview''

It is a review paper discussing the topic of TLRs in the case of Human Immunodeficiency Virus Type 1.

The review paper was carefully prepared. its readability and reception is very clear. The authors did a very good job.

The work should be published as much as possible, but before it is made public, I would suggest a few minor remarks, which I put below:

1. Due to the fact that the authors have prepared a literature review, I would suggest including information on how to select literature for publication. The authors may suggest the PRISMA protocol. The point here is to take into account the inclusion or exclusion criteria of individual publications included in the review.

2. I would suggest expanding the Conclusion section with more information on future perspectives, clinical issues.

Once again, congratulations to the authors of a good review paper.

Author Response

Article titled: ''Toll-like Receptor Response To Human Immunodeficiency Virus Type 1 or Co-infection With Hepatitis B or C virus Infection: An Overview''

It is a review paper discussing the topic of TLRs in the case of Human Immunodeficiency Virus Type 1.

The review paper was carefully prepared. its readability and reception is very clear. The authors did a very good job.

Response: We are very grateful to the reviewer for the compliments.

The work should be published as much as possible, but before it is made public, I would suggest a few minor remarks, which I put below:

1. Due to the fact that the authors have prepared a literature review, I would suggest including information on how to select literature for publication. The authors may suggest the PRISMA protocol. The point here is to take into account the inclusion or exclusion criteria of individual publications included in the review.

Response: We thank the reviewer for the comments. As we did not write a systematic review paper, PRISMA protocol was not followed.

2. I would suggest expanding the Conclusion section with more information on future perspectives, clinical issues.

Response: In response to reviewer comment, we have added future perspectives including clinical issues in the conclusion section (page 14, lines 474-486).

Once again, congratulations to the authors of a good review paper.

Response: Many thanks to the reviewer for the compliments.

Round 2

Reviewer 1 Report

Accept

Reviewer 2 Report

No comments.